# Impact of preventive chemotherapy on *Strongyloides stercoralis*: A systematic review and meta-analysis

Giacomo Stroffolini[1]*, Francesca Tamarozzi[1], Andrea Fittipaldo[1], Cristina Mazzi[1], Brandon Le[2], Susana Vaz Nery[2], Dora Buonfrate[1]

**1** Department of Infectious-Tropical Diseases and Microbiology, IRCCS Sacro Cuore Don Calabria Hospital, Via Don A. Sempreboni, 5, 37024 Negrar, Verona, Italy, **2** The Kirby Institute, University of New South Wales, Sydney, New South Wales, Australia

* giacomo.stroffolini@sacrocuore.it

**Data Availability Statement:** The study protocol was registered in PROSPERO (record

## Abstract

### Background

*Strongyloides stercoralis* is a neglected soil-transmitted helminth (STH) that leads to significant morbidity in endemic populations. Infection with this helminth has recently been recognised by the World Health Organization (WHO) as a major global health problem to be addressed with ivermectin preventive chemotherapy, and therefore, there is now, the need to develop guidelines for strongyloidiasis control that can be implemented by endemic countries. This study aimed to evaluate the impact of ivermectin preventive chemotherapy (PC) on *S. stercoralis* prevalence in endemic areas to generate evidence that can inform global health policy.

### Methodology/Principal findings

This study was a systematic review and meta-analysis. We searched PubMed, EMBASE, Cochrane Central Register of Controlled Trials, and LILACS for literature published between 1990 and 2022 and reporting prevalence of *S. stercoralis* before and after PC with ivermectin, administered either at school or at community level. The search strategy identified 933 records, eight of which were included in the meta-analysis. Data extraction and quality assessment were carried out by two authors. Meta-analysis of studies based on fecal testing demonstrated a significant reduction of *S. stercoralis* prevalence after PC: prevalence Risk Ratio (RR) 0.18 (95% CI 0.14–0.23), $I^2$ = 0. A similar trend was observed in studies that used serology for diagnosis: RR 0.35 (95% CI 0.26–0.48), $I^2$ = 4.25%. A sensitivity analysis was carried out for fecal tests where low quality studies were removed, confirming a post-intervention reduction in prevalence. The impact of PC could not be evaluated at different time points or comparing annual vs biannual administration due to insufficient data.

### Conclusions/Significance

Our findings demonstrate a significant decrease of *S. stercoralis* prevalence in areas where ivermectin PC has taken place, supporting the use of ivermectin PC in endemic areas.

CRD42022355118). All data are available from the manuscript.

**Funding:** This work was supported by the Italian Ministry of Health "Fondi Ricerca Corrente" to IRCCS Sacro Cuore Don Calabria Hospital – L3P1. The funders had no role in study design, data collection and analysis, decision to publish, or preparation of the manuscript.

**Competing interests:** The authors have declared that no competing interests exist.

## Author summary

Ivermectin is a broad-spectrum, anti-parasitic drug that has been administered for decades in the context of preventive chemotherapy programs for onchocerciasis and lymphatic filariasis elimination. Ivermectin is also effective against *Strongyloides stercoralis*, an intestinal worm causing significant morbidity in low-income countries in tropical and subtropical regions. The World Health Organization recently identified the need to develop guidelines to assist *S. stercoralis* control in endemic countries. We aimed to study the impact of ivermectin preventive chemotherapy on *S. stercoralis* prevalence to generate evidence that can inform global health policy. Therefore, we conducted a systematic review of the available literature, collecting data from studies where ivermectin was administered for control of *S. stercoralis* or other parasitic infections. The analysis demonstrated a reduction in the prevalence of *S. stercoralis* in areas where ivermectin was administered. These data may be useful for informing future public health policies.

## Background

*Strongyloides stercoralis* is a soil-transmitted helminth (STH) with a worldwide geographical distribution, estimated to infect more than 600 million people globally, disproportionately in resource-poor communities in tropical and subtropical countries [1]. Although commonly asymptomatic, strongyloidiasis may be associated with eosinophilia, fatigue, and diarrhoea and is distinguished by its auto-infective life cycle, causing chronic infection and potentially fatal hyperinfection in immunosuppressed patients [2]. The disease has recently been recognised by the World Health Organization (WHO) as a major global health problem with the 2030 WHO road map for Neglected Tropical Diseases (NTDs) identifying the need for the development of morbidity control guidelines using ivermectin preventive chemotherapy (PC) [3]. PC, the large-scale delivery of anthelminthic drugs to at-risk populations, is the WHO recommended strategy for control of the other STH species (*Ascaris lumbricoides*, hookworm, *Trichuris trichiura*) [4]. A major limitation of this approach is that the drugs administered for these STH species (albendazole or mebendazole) have low efficacy against *S. stercoralis* [5].

Ivermectin mass drug administration (MDA) has been used for decades in programs for onchocerciasis and lymphatic filariasis (LF) elimination, demonstrating an excellent safety profile (except where *Loa loa* is co-endemic), and high effectiveness against the target filarial parasites [6,7]. Ivermectin MDA is also increasingly being adopted for scabies control [8]. Although ivermectin demonstrated good therapeutic efficacy against *S. stercoralis* in randomized controlled trials, the impact of ivermectin PC on the prevalence of *S. stercoralis* is not completely established [5].

The aim of this systematic review and meta-analysis is to evaluate the impact of ivermectin PC on *S. stercoralis* prevalence in endemic areas with a view to generating evidence that can inform global health policy.

## Materials & methods

### Literature search strategy

The study protocol was registered in PROSPERO (record CRD42022355118). The literature search was conducted on August 11^{th}, 2022 in the following databases: PubMed (MEDLINE), EMBASE, Cochrane Central Register of Controlled Trials, and LILACS. The detailed search strategy is reported in S1 File. The search was restricted to human studies and papers published

between 1990 and 11[th] August 2022, with no language restriction. The search results were combined and duplicates removed before screening for relevance by title and abstract. The reference lists of all included studies as well as of retrieved reviews were searched for other potentially relevant articles. Furthermore, the authors made available, data which were still unpublished at the time of analysis but have been accepted for publication at the time of submission of this manuscript.

The work is presented according to the recommendations of the Preferred Reporting Items for Systematic Reviews and Meta-Analyses (PRISMA) [9]. The PRISMA checklist is provided as S1 File.

## Population, inclusion and exclusion criteria, study design, and outcomes

The target populations considered were school-age children (SAC) and/or adults living in *S. stercoralis* endemic areas and included in ivermectin PC programs targeting specifically *S. stercoralis* or implemented for the control of other parasitic diseases, such as LF or onchocerciasis. Eligible studies were prospective and retrospective longitudinal cohort studies or population-based interventions. Additional inclusion criteria were: human studies published in peer-reviewed journals; publications presenting original data; and original data obtainable from full text or abstract. Exclusion criteria were: individually randomized controlled trials (RCTs) or publications with other study designs assessing the efficacy of ivermectin for individual treatment; and reviews and conference abstracts.

## Study selection and data extraction

Rayyan software [10] was used to screen records obtained with the search strategy described above. The titles and abstracts of identified articles were screened for potential eligibility in parallel by two authors [FT & GS]; then, full texts of potentially eligible articles were reviewed. At all steps, disagreements between authors were resolved by discussion, with the support of a third senior author [DB], if consensus could not be reached. Data were extracted in parallel by the same two authors using a Microsoft Excel (Version 2019) spreadsheet. The following data were extracted: study design; study population; country where the study was conducted; primary parasitic target(s) of the intervention; schedule of the intervention (ivermectin dose, schedule of administration, number of rounds of ivermectin distribution, duration of the intervention); entire study duration (from first intervention to last follow-up time point); water, sanitation and hygiene (WASH) interventions implemented during the study period; number of participants recruited at baseline (intervention and no-intervention groups in case of the inclusion of a non-intervention area in the study design); number of participants evaluated and number of participants positive for *S. stercoralis* infection at baseline and at last follow-up timepoint by each diagnostic technique (formol-ether concentration [FECT], direct fecal smears, Baermann, agar plate culture [APC] or other culture methods, PCR, serology); *S. stercoralis* prevalence point estimate and 95% confidence interval (CI) at baseline and at last follow-up time point by each diagnostic technique in case absolute numbers were not provided; and length of time between last treatment and last follow-up evaluation.

For the purpose of the analysis, data obtained with diagnostic techniques applied on feces (FECT, direct fecal smear, Baermann, APC or other culture methods, and PCR) were grouped together ("fecal tests"). Similarly, data obtained from different serological methods/tests were grouped together ("serology").

## Quality assessment

The same two authors independently assessed the quality of studies based on an adapted version of the Newcastle-Ottawa Scale [11] (S2 File). The studies were evaluated in the following domains: selection (items: study design, representativeness of the exposed cohort; technique(s) used to ascertain the infection; data availability); comparability (items: presence of a control cohort, consistency of diagnostic methods applied at baseline and follow-up); and outcome (items: blinding, appropriateness of follow-up length, representativeness of the follow-up cohort). A maximum of one "star" (*) per item was assigned, resulting in studies scoring 8–9* classified as very high quality, 5–7* as high quality, and 4* or less as low quality. S2 File summarizes the items included in each domain and the characteristics for the attribution of the "*" for each item (scoring system).

## Statistical analysis and data synthesis

The primary outcome was the change in prevalence of *S. stercoralis* between baseline and the last timepoint assessed after the last round of administration of ivermectin. The secondary outcome was comparison of the impact (prevalence before and after the intervention) of different schedules of intervention (e.g., annual versus biannual treatment), if sufficient data were available.

For all outcomes, random-effects models were used to calculate pooled prevalence risk ratios (RR) and 95% confidence intervals of the change in *S. stercoralis* prevalence between baseline and follow-up. The outcome variable in the model was the prevalence reduction, **calculated by dividing the positivity rate at follow-up by the positivity rate at baseline.** RRs equal to 1 indicate no change in prevalence between baseline and follow-up, RRs <1 indicate a relative decrease in prevalence between baseline and follow-up, and RRs >1 indicate an increase in prevalence between baseline and follow-up. Sensitivity analyses were performed considering the quality of the studies and the number of rounds of ivermectin administered. Heterogeneity between studies was assessed with $I^2$ statistics. $I^2$ values of 50% or more were considered heterogeneous [12]. A P value of less than 0.05 was considered significant. All statistical analyses were conducted using Stata version 17.0 (Stata Corp, College Station, TX, USA).

# Results

Of the 933 publications identified by electronic search after duplicate removal, 8 (including the one provided by the authors) were included in the final analysis (Fig 1). Main characteristics of the included articles are shown in Table 1.

Ivermectin was administered at a 200 µg/kg single dose in all studies. In all studies, ivermectin was administered both to SAC and adults, but prevalence data were sometimes reported for either SAC or adults only. The median duration of the intervention period (defined as time between first and last round of PC) was 24 months [1–192, Min-Max], and the median duration of the studies (including last follow-up) was 27 months; assessment of prevalence of strongyloidiasis was carried out at a median of 12 months after the last round of ivermectin (6–84, Min-Max). No concomitant WASH interventions were disclosed in any of the retrieved studies. No study included a non-intervention area, with the exception of that by Anselmi et al [13], who reported *S. stercoralis* prevalence both in communities where ivermectin was massively distributed (in the context of the onchocerciasis elimination program) and in those that did not receive the intervention. The first assessment of *S. stercoralis* prevalence in the no-intervention area was carried out five years after the start of ivermectin distribution in the intervention area. Furthermore, some other interventions carried out independently from the

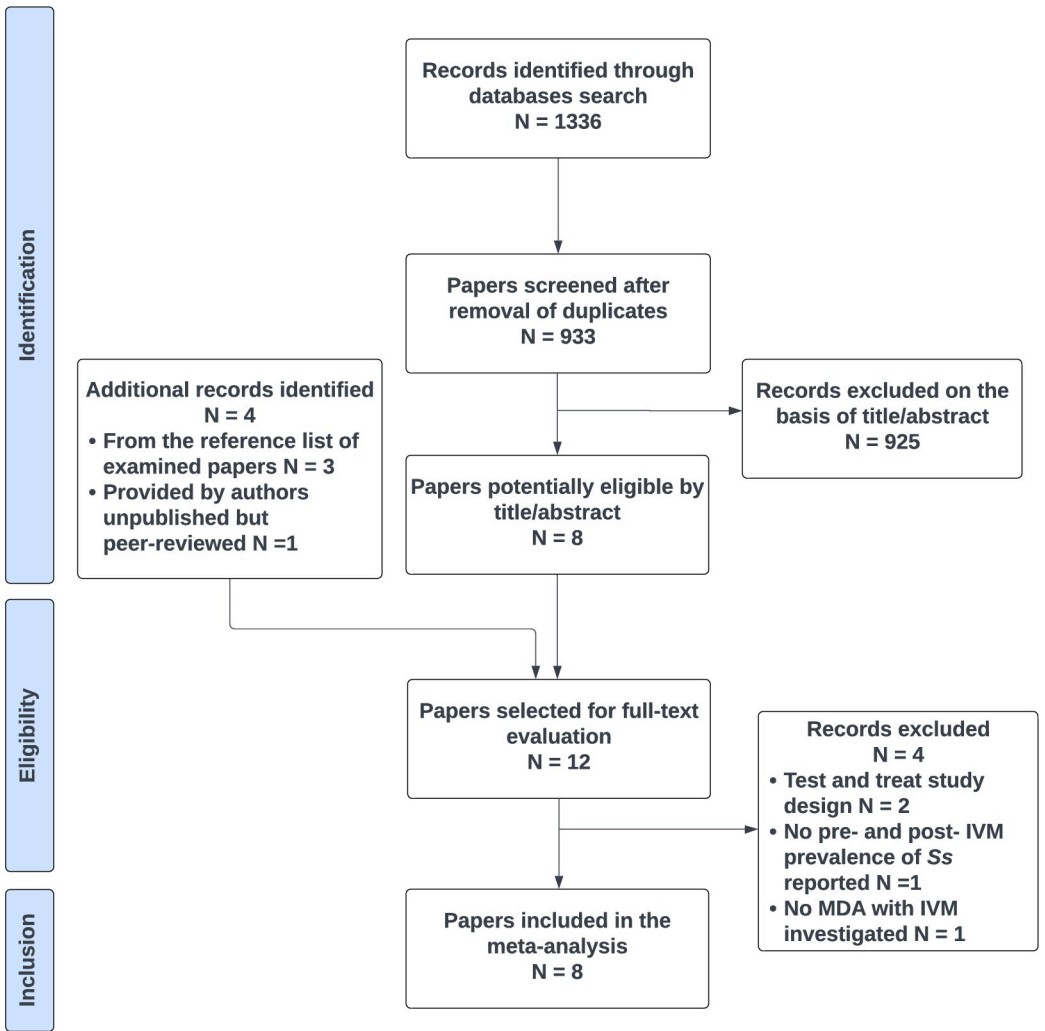

**Fig 1. PRISMA flow diagram.**

distribution of ivermectin for onchocerciasis (e.g., sanitary improvements) could have influenced the prevalence of *S. stercoralis* infection in a not homogeneous manner between the areas. Nevertheless, all this considered, it must be noted that at the last evaluation carried out six years after the end of ivermectin distribution, *S. stercoralis* prevalence in the intervention area (assessed by a combination of fecal and serology tests) was 3% in adults and 1% in children, while in the no-intervention area was 18% in adults and 9% in children.

Meta-analysis of the six studies using feces to detect infection with *S. stercoralis* demonstrated a significant reduction of *S. stercoralis* prevalence after PC (Fig 2), with a RR of 0.18 (95% CI 0.14–0.23), indicating a relative reduction of 82%; heterogeneity was assessed as $I^2 = 0$. A similar trend was found in the meta-analysis of the three studies using serology (Fig 3), with a RR of 0.35 (95% CI 0.26–0.48), thus indicating a relative reduction of 65%. Although three studies only were available in the serology subgroup analysis, the heterogeneity between them was low ($I^2 = 4.25\%$).

A sensitivity analysis was carried out for studies based on fecal tests, based on their quality score. The meta-analysis of the two high quality studies (Fig 4) showed a reduction in post-PC prevalence of *S. stercoralis* (RR 0.22, 95% CI 0.02–2.25) despite their high heterogeneity ($I^2 = $

**Table 1. Main characteristics of included papers.**

| Study | Country | Target population for which data were extracted | Target of the intervention | Drug distributed in addition to IVM | IVM schedule | Total n of IVM rounds administered | Duration of the entire study (in months from baseline assessment to last follow up) | Length of follow-up (in months) after last IVM distribution | Tested population at baseline and last follow-up | Ss prevalence at baseline and last follow-up | Diagnostic tests used at baseline and last follow-up | Quality |
|---|---|---|---|---|---|---|---|---|---|---|---|---|
| Anselmi et al. (2015) [13] | Ecuador | Adults § | *Onchocerca volvulus* | None | Once a year for 10 years then twice a year for 7 years | 24 | 276 | 72 | BL = 118 FU = 150 | BL = 6.8% FU = 0.7% | BL = direct fecal smear FU = FECT | 4* (low) |
| Barda et al. (2017) [14] | Tanzania | SAC§ | *Wuchereria bancrofti* | PZQ +ABZ | Once a year | 6 | 156 | 84 | BL = 525 FU = 440 | BL = 41% FU = 7% | BL = Harada-Mori culture FU = APC +Baermann | 3* (low) |
| Echazu et al. (2017)[15] | Argentina | Adults and SAC | STH including Ss | ABZ | Intervals between IVM 9–16 months | 3 | 36 | 12 | BL = 1982 FU = 2685 | BL serology = 51% BL fecal = 16,8% FU serology = 13,7% FU fecal = 3.3% | BL and FU fecal = Harada-Mori culture +APC +Baermann BL and FU serology = NIE ELISA | 4* (low) |
| Heukelbach et al. (2004) [16] | Brazil | Adults and SAC | STH including Ss | None | Single round, composed by 2 doses (10 days apart) | 1 | 9 | 9 | BL = 516 FU = 403 | BL = 11% FU = 0.7% | BL and FU = Baermann | 5* (high) |
| Kearns et al. (2017)[17] | Australia | Adults and SAC | Ss | None | Once a year | 2 | 18 | 6 | BL = 1013 FU = 238 | BL = 21% FU = 17% | BL and FU = in house ELISA based on *S. ratti* | 5* (high) |
| Knopp et al. (2009)[18] | Tanzania | SAC§ | *Wuchereria bancrofti* | MBZ/ABZ and PZQ in children | Once a year | 6 | 78 | 6 | BL = 1204 FU = 364 | BL = 34.8% FU = 6.6% | BL = Baermann FU = APC +Baermann | 4* (low) |
| Marks et al. (2020)[19] | Solomon Islands | SAC§ | Scabies | None | Single round, composed by 2 doses (10 days apart, if scabies positive) | 1 | 12 | 12 | BL = 260 FU = 223 | BL = 15.8% FU = 6.7% | BL and FU = NIE-based fluorescent bead assay | 7* (high) |

*(Continued)*

**Table 1.** (Continued)

| Study | Country | Target population for which data were extracted | Target of the intervention | Drug distributed in addition to IVM | IVM schedule | Total n of IVM rounds administered | Duration of the entire study (in months from baseline assessment to last follow up) | Length of follow-up (in months) after last IVM distribution | Tested population at baseline and last follow-up | Ss prevalence at baseline and last follow-up | Diagnostic tests used at baseline and last follow-up | Quality |
|---|---|---|---|---|---|---|---|---|---|---|---|---|
| Le *et al.* (2023)[20] | Timor-Leste | SAC | *Wuchereria bancrofti*, scabies, STH including Ss | DEC+ABZ | Once a year | 1 | 18 | 18 | BL = 1185 FU = 1196 | BL = 1.1% FU = 0.8% | BL and FU = PCR | 7* (high) |

**Key:** BL: baseline; FU: follow-up; IVM: ivermectin; MBZ: mebendazole; ABZ: albendazole; PZQ: praziquantel; Ss: *Strongyloides stercoralis*; APC: agar plate culture; FECT: formol-ether concentration technique; SAC: school-age children; § Intervention carried out in both children and adults, but data for analysis extractable only for the adults group.

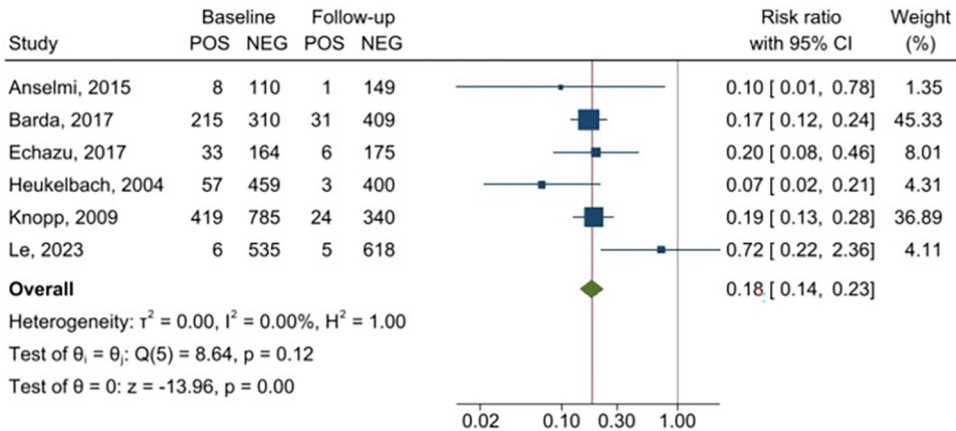

**Fig 2. Forest plot displaying *Strongyloides stercoralis* prevalence before and after the intervention (assessment with fecal tests).**

87.4%). Similarly, the meta-analysis of the four studies of low-quality mirrors the main results (RR = 0.18, 95% CI 0.14–0.23). The sensitivity analysis of the studies using fecal tests according to the number of ivermectin rounds (single versus multiple rounds) is shown in Fig 5. Meta-analysis of the two studies assessing the impact of a single round of ivermectin showed a significant reduction of *S. stercoralis* prevalence (RR = 0.22, 95% CI 0.02–2.25) although high heterogeneity ($I^2$ = 87.41%) of the studies should be considered. A similar trend was observed for the meta-analysis of the four studies that conducted multiple rounds of ivermectin PC, comparing baseline prevalence and prevalence at the last follow-up after two or more rounds of ivermectin (RR = 0.18, 95% CI 0.14–0.23). The effect of PC could not be evaluated comparing annual vs biannual administration scheme due to insufficient data. Sensitivity analysis could not be done for studies using serology due to insufficient data.

## Discussion

To our knowledge, this study is the first to systematically evaluate the available evidence on the impact of ivermectin PC on *S. stercoralis* prevalence. Our findings demonstrate a significant decrease in *S. stercoralis* prevalence in areas where ivermectin PC has taken place, supporting

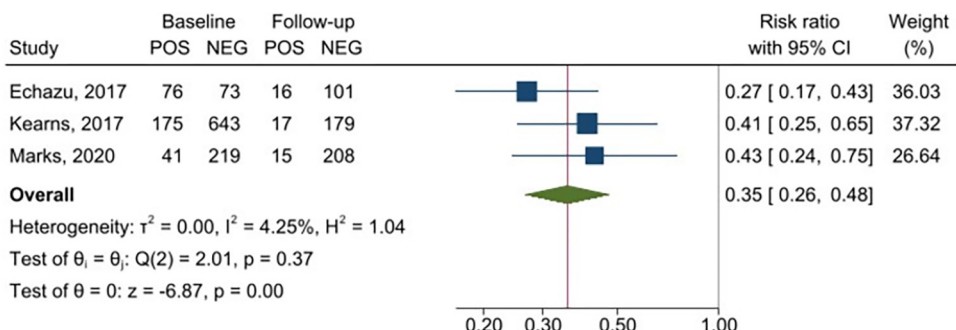

**Fig 3. Forest plot displaying *Strongyloides stercoralis* prevalence before and after the intervention (assessment with serology).**

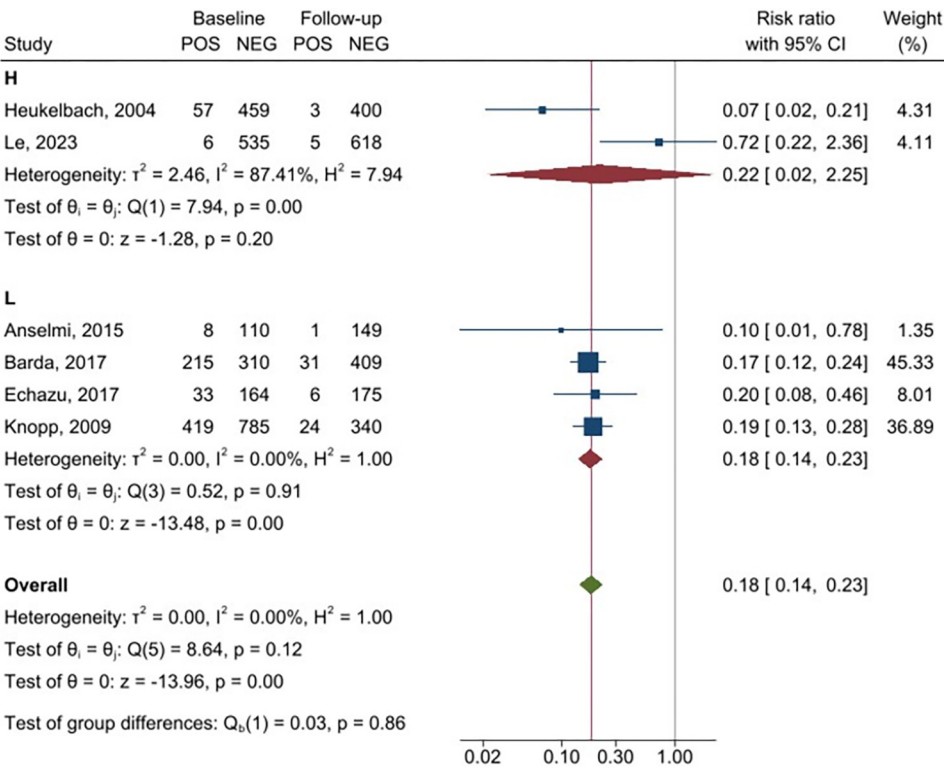

**Fig 4. Forest plot displaying *Strongyloides stercoralis* prevalence before and after the intervention (assessment with fecal tests), stratified by quality of the studies.**

its use as a public health strategy for strongyloidiasis control. This finding was consistent in our subgroup analysis across different diagnostic techniques. The difference in the relative reduction of *S. stercoralis* prevalence shown by the meta-analysis of studies based on fecal tests (82%) compared to those based on serology (65%) is consistent with the different characteristics of the diagnostic tests. Fecal tests are usually less sensitive than serological assays [21], hence might tend to overestimate the efficacy of the intervention; moreover, they do turn negative in a shorter timeframe compared to serology, that often requires more than 12 months for seroconversion [21]. Hence, the effectiveness of ivermectin PC on *S. stercoralis* reduction might be better considered in a range between 65% and 82%.

Only the paper by Anselmi et al [13] included a no-intervention area. Data from this area could not be formally analyzed and compared with the intervention area since baseline prevalence of infection was not evaluated at the same time as it was in the treatment area (i.e. before start of ivermectin distribution). However, these results further support the effectiveness of ivermectin PC implemented at whole-population level.

Currently, the 2021–2030 WHO road map for NTDs identifies the need to implement ivermectin PC programs for *S. stercoralis* control targeting school-age children [3]. However, a detailed strategy is still under development, and specific indications regarding the infection prevalence threshold to begin PC, the number and frequency of rounds of ivermectin, and other possible practical recommendations will have to be defined on the basis of epidemiological and implementation factors, as well as control targets.

Ivermectin MDA has been a safe and highly effective tool for the control and elimination of LF and onchocerciasis as a public health problem for the last two decades, and is increasingly

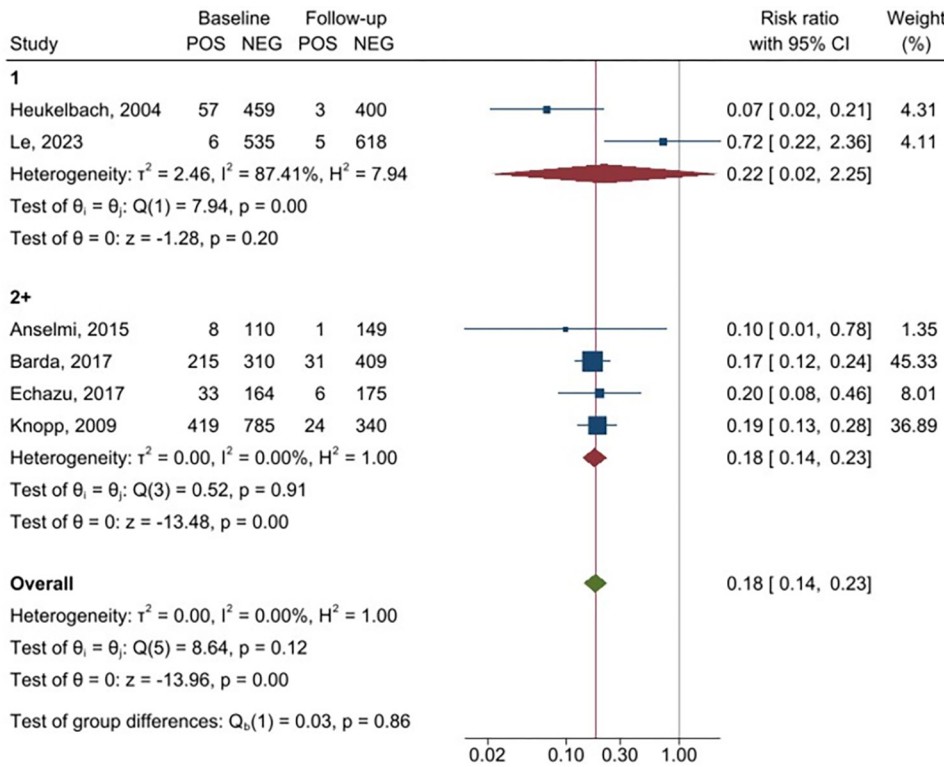

**Fig 5. Forest plot displaying *Strongyloides stercoralis* prevalence before and after the intervention (assessment with fecal tests), stratified by number of ivermectin rounds.**

being used for scabies control [6,7,8]. Although ivermectin is currently contraindicated in children less than 5 years old due to safety concerns, there is burgeoning evidence that the drug is safe also in this group [22]. We hope that investments would be made for pediatric formulations and registration, to allow younger children benefit from treatment against strongyloidiasis.

Currently, ivermectin is donated only for the elimination programs for LF and onchocerciasis [23]. The WHO prequalification of two generic ivermectin products is a step in that direction. Our evidence may be used as additional proof to make ivermectin available at lower cost, so that it is accessible to low-middle income counties and also outside programs for these filarial infections. Furthermore, the integration of PC targeting *S. stercoralis* in the ongoing STH control programs, using albendazole and ivermectin, would enable the control of all STH species. However, additional costs would have to be envisaged due to the need for the use of diagnostic tests different from those routinely used for the other STH, such as Kato-Katz, which are inadequate for *S. stercoralis* when conducting baseline and impact assessment surveys [21].

Limitations of this study should be considered. First, this review was limited by the small number of studies included, while the data available did not allow a number of analyses that could have been informative for the implementation of PC for *S. stercoralis*. For instance, we could not explore the impact of school-based PC on *S. stercoralis* prevalence, as all included studies reported data on community-wide ivermectin administration. It is in fact, possible that PC targeting SAC only, as envisaged for the other STH, might be less effective in reducing the prevalence of strongyloidiasis compared to community-based administration. Further studies are needed to confirm an effective reduction in prevalence of *S. stercoralis* following school-

based PC. Another analysis that could not be carried out concerned differences in follow-up time periods; this, combined with the high heterogeneity of the sensitivity analysis of studies reporting prevalence after a single round of PC, did not permit drawing conclusions about the effectiveness of PC over time. Second, in most studies, the target of PC was not *S. stercoralis*, and data on prevalence of this parasite were retrospectively retrieved, thus limiting our ability to accurately quantify the true impact of PC on *S. stercoralis*. Finally, there was considerable variation in the type of diagnostic test used, between studies; for instance, fecal tests included methods with variable sensitivity ranges (from 0–18% for a single direct smear to 60–98% for Baermann) [21]; however, both meta-analyses grouping fecal tests on one side and serology assays on the other, showed low heterogeneity, hence results can be deemed reliable.

## Conclusions

This study demonstrates that ivermectin PC leads to a significant reduction in prevalence of *S. stercoralis* in endemic areas, supporting recent WHO recommendations. Several implementation parameters still need to be defined to ensure effective control of *S. stercoralis*, such as the threshold prevalence to start PC and diagnostic methods to define it, population targets (SAC versus entire communities), target prevalence that would allow control programs to stop, and the schedule of ivermectin distribution.

## Supporting information

**S1 File. PRISMA checklist.**
(TIF)

**S2 File. Modified Newcastle-Ottawa Scale.**
(PDF)

## Author Contributions

**Conceptualization:** Giacomo Stroffolini, Francesca Tamarozzi, Dora Buonfrate.

**Data curation:** Giacomo Stroffolini, Francesca Tamarozzi, Brandon Le, Susana Vaz Nery, Dora Buonfrate.

**Formal analysis:** Andrea Fittipaldo, Cristina Mazzi, Dora Buonfrate.

**Funding acquisition:** Cristina Mazzi, Dora Buonfrate.

**Investigation:** Andrea Fittipaldo, Cristina Mazzi, Dora Buonfrate.

**Methodology:** Andrea Fittipaldo, Cristina Mazzi, Dora Buonfrate.

**Project administration:** Cristina Mazzi, Dora Buonfrate.

**Resources:** Cristina Mazzi, Dora Buonfrate.

**Software:** Cristina Mazzi, Dora Buonfrate.

**Validation:** Cristina Mazzi, Dora Buonfrate.

**Visualization:** Cristina Mazzi, Dora Buonfrate.

**Writing – original draft:** Giacomo Stroffolini, Francesca Tamarozzi, Andrea Fittipaldo, Cristina Mazzi, Brandon Le, Susana Vaz Nery, Dora Buonfrate.

**Writing – review & editing:** Giacomo Stroffolini, Francesca Tamarozzi, Andrea Fittipaldo, Cristina Mazzi, Brandon Le, Susana Vaz Nery, Dora Buonfrate.

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
