## [Decision Letter · Decision Letter 0]

16 May 2023

Dear Dr Stroffolini,

Thank you very much for submitting your manuscript "Impact of preventive chemotherapy on Strongyloides stercoralis: a systematic review and meta-analysis." for consideration at PLOS Neglected Tropical Diseases. As with all papers reviewed by the journal, your manuscript was reviewed by members of the editorial board and by several independent reviewers. In light of the reviews (below this email), we would like to invite the resubmission of a significantly-revised version that takes into account the reviewers' comments. 

We cannot make any decision about publication until we have seen the revised manuscript and your response to the reviewers' comments. Your revised manuscript is also likely to be sent to reviewers for further evaluation.

Sincerely,

De'Broski R Herbert

Academic Editor

Eva Clark

Section Editor

Reviewer's Responses to Questions

**Key Review Criteria Required for Acceptance?**

**Methods**

-Are the objectives of the study clearly articulated with a clear testable hypothesis stated?

-Is the study design appropriate to address the stated objectives?

-Is the population clearly described and appropriate for the hypothesis being tested?

-Is the sample size sufficient to ensure adequate power to address the hypothesis being tested?

-Were correct statistical analysis used to support conclusions?

-Are there concerns about ethical or regulatory requirements being met?

Reviewer #1: The objective of the review and meta-analysis was clearly articulated, and the study design was appropriate to address this. 

The inclusion and exclusion criteria for studies included in the review and meta-analysis were clearly stated and were appropriate for the study.

The sample size, as shown by the small number of studies included in the review meta-analysis, was not sufficient. However, this limitation was beyond the authors, as data which accounted for the limitation, were not available in included studies. This limitation was stated by the authors in their conclusion. 

The conclusions were backed up by correct statistical analyses

There are no concerns about ethical or regulatory requirements being met for the conduct of the study

Reviewer #2: -Are the objectives of the study clearly articulated with a clear testable hypothesis stated? The goal of the paper should be reviewed since there is not enough information in the literature to actually support the intention of the paper.

-Is the study design appropriate to address the stated objectives? Yes

-Is the population clearly described and appropriate for the hypothesis being tested? Yes

-Is the sample size sufficient to ensure adequate power to address the hypothesis being tested? No

-Were correct statistical analysis used to support conclusions? Yes

-Are there concerns about ethical or regulatory requirements being met? No

Reviewer #3: Methods employed are adequate.

**Results**

-Does the analysis presented match the analysis plan?

-Are the results clearly and completely presented?

-Are the figures (Tables, Images) of sufficient quality for clarity?

Reviewer #1: The analysis presented match the analysis plan.

The results were clearly presented.

However,:

1. Year of publication should follow each of the author names for all studies listed under the column Study" in Table 1.

2. et al. should follow each of the author names listed under the column "Study" in Figures 2, 3, 4 & 5.

Reviewer #2: -Does the analysis presented match the analysis plan? Yes

-Are the results clearly and completely presented? In part. There is a substantial amount of information in the papers that they use as a reference that could be explored to increase the relevance of the review. 

-Are the figures (Tables, Images) of sufficient quality for clarity? Yes

Reviewer #3: Results are adequately presented and dicussed.

**Conclusions**

-Are the conclusions supported by the data presented?

-Are the limitations of analysis clearly described?

-Do the authors discuss how these data can be helpful to advance our understanding of the topic under study?

-Is public health relevance addressed?

Reviewer #1: The conclusions are justified by the data presented in the manuscript.

The limitations of the analysis were described, although not in the conclusion, but rather in the "Discussion" section.

The authors did discuss how data from their study could be helpful to advance our understanding of the subject matter under study as well as the public health relevance of their study/analysis. These were captured in the "Discussion" section.

Reviewer #2: -Are the conclusions supported by the data presented? Yes, but since the authors don't have enough information to support their main goal, the conclusion is scarce.

-Are the limitations of analysis clearly described? Yes

-Do the authors discuss how these data can be helpful to advance our understanding of the topic under study? Yes

-Is public health relevance addressed? Yes

Additional comment: The paragraph were the authors discuss the use of a combination of anthelminthics is not helpful to prove their point, and makes the reader wonder why they don't did the review also based in other possible treatments.

Reviewer #3: Conclusions are adequately formulated and limitations in the analysis are clearly detailed. The public health implications of the findings are addressed.

**Editorial and Data Presentation Modifications?**

Reviewer #1: Minor Revision; mainly editorial.

Please see attached "Reviewed" manuscript for details.

Reviewer #2: (No Response)

Reviewer #3: Reference [3] is not the WHO road map for NTDs, although the text on lines 49-51 seems to imply this. You may wish to add the road map as appropriate.

Please spell "roadmap" as "road map" (two words).

**Summary and General Comments**

Reviewer #1: The conceptualization and execution of the study was adequate. 

The "References" section needs to be adequately revised. Please see attached "Reviewed" manuscript.

Reviewer #2: As addressed by the authors in the discussion there is not enough data in the literature that support this review. With that, there is no clear conclusion and no addition of information to the knowledge that the scientific community already has.

Reviewer #3: An interesting review assessing the impact of preventive chemotherapy (PC) with ivermectin on prevalence of S. stercoralis infection in endemic areas, with the aim of generating evidence that can inform global health policy.

In spite of the limited number of studies included in the analysis and the heterogeneity of such studies in terms of target population, frequency of administration of IVM and follow-up delay, results are consistently supporting MDA with IVM as a key public health intervention against Strongyloides stercoralis.

I don't have any specific technical comment on the article.

PLOS authors have the option to publish the peer review history of their article (what does this mean?). If published, this will include your full peer review and any attached files.

Reviewer #1: No

Reviewer #2: No

Reviewer #3: No
---

## [Decision Letter · Decision Letter 1]

23 Jun 2023

Dear Dr. Stroffolini,

We are pleased to inform you that your manuscript 'Impact of preventive chemotherapy on Strongyloides stercoralis: a systematic review and meta-analysis.' has been provisionally accepted for publication in PLOS Neglected Tropical Diseases.

Best regards,

De'Broski R Herbert

Academic Editor

Eva Clark

Section Editor

Nicely improved

Reviewer's Responses to Questions

**Key Review Criteria Required for Acceptance?**

**Methods**

-Are the objectives of the study clearly articulated with a clear testable hypothesis stated?

-Is the study design appropriate to address the stated objectives?

-Is the population clearly described and appropriate for the hypothesis being tested?

-Is the sample size sufficient to ensure adequate power to address the hypothesis being tested?

-Were correct statistical analysis used to support conclusions?

-Are there concerns about ethical or regulatory requirements being met?

Reviewer #1: The objective of the review and meta-analysis was clearly articulated, and the study design was appropriate to address this.

The inclusion and exclusion criteria for studies included in the review and meta-analysis were clearly stated and were appropriate for the study.

The sample size, as shown by the small number of studies included in the review meta-analysis, was not sufficient. However, this limitation was beyond the authors, as data which accounted for the limitation, were not available in included studies. This limitation was stated by the authors in their conclusion.

The conclusions were backed up by correct statistical analyses

There are no concerns about ethical or regulatory requirements being met for the conduct of the study.

Reviewer #3: Methods are adequate and adequately described

**Results**

-Does the analysis presented match the analysis plan?

-Are the results clearly and completely presented?

-Are the figures (Tables, Images) of sufficient quality for clarity?

Reviewer #1: The analysis presented match the analysis plan.

The results were clearly presented.

However,:

1. All author names in Figures 2-5 should be followed by et al. which precedes the year of publication in parentheses, e.g., Anselmi et al. (2015), Le et al. (2023).

Reviewer #3: Results are adequately presented

**Conclusions**

-Are the conclusions supported by the data presented?

-Are the limitations of analysis clearly described?

-Do the authors discuss how these data can be helpful to advance our understanding of the topic under study?

-Is public health relevance addressed?

Reviewer #1: The conclusions are justified by the data presented in the manuscript.

The limitations of the analysis were described, although not in the conclusion, but rather in the "Discussion" section.

The authors did discuss how data from their study could be helpful to advance our understanding of the subject matter under study as well as the public health relevance of their study/analysis. These were captured in the "Discussion" section.

Reviewer #3: Conclusion are adequate

**Editorial and Data Presentation Modifications?**

Reviewer #1: Minor Revision; mainly editorial.

Reviewer #3: N/A

**Summary and General Comments**

Reviewer #1: The conceptualization and execution of the study was adequate.

Most of the errors outlined in the original manuscript have now been corrected. However, the minor revision suggested should be corrected to make for a better article. Please see attached "Reviewed" manuscript.

Reviewer #3: The revised version of the manuscript can be published

PLOS authors have the option to publish the peer review history of their article (what does this mean?). If published, this will include your full peer review and any attached files.

Reviewer #1: No

Reviewer #3: No

---

## [Editor Report · Acceptance letter]

3 Jul 2023

Dear Dr Stroffolini,

We are delighted to inform you that your manuscript, "Impact of preventive chemotherapy on Strongyloides stercoralis: a systematic review and meta-analysis.," has been formally accepted for publication in PLOS Neglected Tropical Diseases.

Best regards,

Shaden Kamhawi

co-Editor-in-Chief

Paul Brindley

co-Editor-in-Chief
